# Neurological Manifestation of Canine Distemper Virus: Increased Risk in Young Shih Tzu and Lhasa Apso with Seasonal Prevalence in Autumn [note 1]

**DOI:** 10.3390/v17060820

**Published:** 2025-06-06

**Authors:** Heloisa L. Freire, Ítalo H. N. Iara, Luana S. R. Ribeiro, Paulo A. O. Gonçalves, David H. Matta, Bruno B. J. Torres

**Affiliations:** 1Escola de Veterinária e Zootecnia, Universidade Federal de Goiás, Goiânia 74961-886, GO, Brazil; heloisaloures@discente.ufg.br (H.L.F.); italo_iara@discente.ufg.br (Í.H.N.I.); 2Instituto de Ciência Agrárias, Universidade Federal dos Vales do Jequitinhonha e Mucuri, Unaí 38610-000, MG, Brazil; luana.ribeiro@ufvjm.edu.br; 3Instituto de Matemática e Estatística, Universidade Federal de Goiás, Goiânia 74961-886, GO, Brazil; goncalves2@discente.ufg.br (P.A.O.G.); dhmatta@ufg.br (D.H.M.)

**Keywords:** dogs, infectious diseases, epidemiology, neurology, paramoxyvirus

## Abstract

Canine distemper virus (CDV) is a highly contagious disease with high morbidity and mortality rates in veterinary medicine. This retrospective study aimed to identify epidemiological characteristics and potential risk factors associated with CDV infection in dogs exhibiting neurological manifestations. The diagnosis was confirmed through immunochromatographic antigen testing, RT-PCR, or *Lentz corpuscles* identification. Dogs diagnosed with central nervous system (CNS) disorders unrelated to CDV served as the control group. Age, breed, weight, sex, and neuter status were compared between groups using logistic regression (*p* < 0.05), the log-likelihood method, and log odds ratio (LOR) calculations. Clinical signs, seasonality, and vaccination protocols were documented. Prevalence, mortality, lethality, and survival rates were determined. Younger dogs (*p* = 0.00690; LOR = −0.01438) and Shih Tzu (*p* = 0.00007; LOR = 1.53774) and Lhasa Apso (*p* = 0.000264; LOR = 1.76084) showed a significantly increased likelihood of developing neurological signs due to CDV infection. Most CDV-infected dogs exhibited multifocal CNS involvement and accompanying extra-neural signs. The highest occurrence of CDV-related neurological signs was recorded in autumn. Many infected dogs had an updated vaccination protocol. The prevalence of dogs with CDV was 4.72%. Mortality and lethality rates were 1.94% and 47.06%, respectively. The median survival time was 754 days. The identified epidemiological characteristics and risk factors provide essential insights for improving preventive strategies against CDV infection.

## 1. Introduction

Canine distemper virus (CDV) belongs to the *Paramyxoviridae* family and *Morbillivirus* genus [1,2,3]. It is a highly contagious and deadly pathogen that poses significant threats to domestic dogs and wildlife [4]. CDV is one of the most significant diseases in veterinary neurology due to its high morbidity, mortality, and neurological sequelae in domestic dogs [2,3,4,5]. The virus is transmitted primarily through aerosolized viral particles from infected secretions [6,7].

Clinical signs of CDV infection can include respiratory, gastrointestinal, ophthalmologic, dermatologic, and neurological signs [8,9]. While neurological involvement is common, it is not exclusive to CDV infection [10,11]. Diagnosis is based on clinical history, physical examination, and complementary tests, including blood and urine analysis, cerebrospinal fluid evaluation, immunochromatographic antigen testing, and reverse transcriptase polymerase chain reaction (RT-PCR) [12].

Epidemiological investigations have described the wide range of neural and extra-neural manifestations of CDV. However, the lack of a control group for statistical comparison limits the identification of real risk factors associated with disease development [2,9,10,11]. These considerations highlight the importance of recognizing risk factors to enable a prompt and accurate diagnosis.

Although CDV was traditionally believed to affect dogs regardless of age, sex, weight, or breed [13], several studies have reported contradictory epidemiological findings, suggesting that these factors can influence CDV prevalence. Regarding susceptibility between males and females, most studies show no gender preference [8,14,15,16]. One study found that dogs with good body scores were more likely to test positive for CDV [13]. Breed seems to be a contradictory point, since some studies showed higher risk in mixed-breed dogs [14,16], and others show no influence [13,15,17], but none shows specific breed-related risk. Also, one study showed that dolichocephalic breeds can also be more susceptible to developing encephalitis by CDV than brachycephalic ones [16].

Age, vaccination status, and urbanization are the most common points of accordance in studies. Unvaccinated dogs, or those with incomplete vaccination, are at significantly increased risk and this would make young dogs, especially those under four months of age, highly susceptible [2,13]. Nevertheless, there are studies that animals between one and six years had a higher risk of developing CDV infection [4,8,11,14,15,16,18]. Also, dogs in high-contact environments like kennels and shelters also faced higher chances of exposure [19]. Additionally, exposure to infected wild animals can increase the risk for domestic dogs [7]. However, these studies were conducted without including a control group of negative dogs, limiting the possibility of confirming risk factors with greater statistical power.

Clinical manifestation is also a not well-established point. Acute and chronic presentations, with and without neurological signs, are possible since the virus starts its replication at the respiratory tract and can affect various organs [10,17]. However, the frequency of infected dogs associated with risk factors and the most common clinical manifestations remain poorly characterized.

No specific therapeutic protocol is available for CDV, and approximately 89% of dogs that develop severe clinical signs either die or are euthanized, particularly in cases involving the nervous system [9]. Epidemiological studies can improve disease control and treatment strategies [20].

To the best of the authors’ knowledge, although neurological involvement is a critical aspect of canine distemper, no previous studies have particularly investigated its neurological form using a control group to compare dogs with neurological signs attributable to other etiologies. Specifically, limited data exist on the relationship between patient demographics, such as breed, age, sex, vaccination, and the neurological form of the disease, particularly in contrast with other CNS conditions. Therefore, this study aimed to characterize epidemiological features and potential risk factors associated with the neurological form of canine distemper, using a control group of dogs with other neurological diseases for comparative analysis.

## 2. Materials and Methods

A retrospective analysis of medical records from the Neurology and Neurosurgery Service of the Veterinary Medical Teaching Hospital at the Federal University of Goiás was conducted. Cases were selected from January 2018 to December 2022 and categorized into distemper group (DG) or control group (CG). The DG comprised dogs with neurological signs that were positive for CDV, confirmed via immunochromatographic antigen testing, RT-PCR, and/or the identification of *Lentz corpuscles*. The CG includes dogs with CNS conditions but testing negative for CDV. Dogs with incomplete medical records or exhibiting clinical signs and/or laboratory indications of distemper that were not subjected to molecular testing were excluded from the study.

Data collected of signalment (age, breed, weight, sex, and neutering status) were extracted from the medical records of both groups. Neurological signs in DG cases were categorized based on neurolocalization (forebrain, brainstem, cerebellum, spinal cord, or multifocal). Extra-neural signs were also recorded. The elapsed time from the onset of neurological signs to the presentation was recorded. Additionally, the temporal relationship between the onset of systemic and neurological signs was assessed, expressed in days, and categorized as occurring before, simultaneously, or after. The phase of the disease was defined as acute if the onset of neurological signs was at a maximum 14 days before the presentation, or chronic if the neurological signs had the onset at 15 days or more. Seasonal trends (month, year, and season) were analyzed based on the onset of clinical signs. Finally, vaccination protocols for core vaccines, as recommended by WSAVA [21], were reviewed. The protocol was classified as either updated (prime vaccination and annual booster) or outdated (absence of prime vaccination or annual booster) as usually practiced in Latin America [22]. Also, vaccine brand and if it was performed by a qualified professional were recorded.

The prevalence of CDV was determined by calculating the ratio of the number of dogs with a confirmed diagnosis to the total population of interest that visited the veterinary service during the study period, restricted to individuals who met the predefined inclusion criteria. Mortality was calculated by dividing the number of animals that died by the total number of animals included in the study. Lethality was calculated by dividing the number of dogs that died by the total number of infected animals (DG). The reason for death, disease phase (acute or chronic), and whether death occurred naturally or by euthanasia were recorded. Kaplan–Meier survival curve was also calculated using 19 October 2023 as the last day of contact with the owners.

The results were subjected to logistic regression analysis using R software (Version 4.4.2, R Core Team, 2024), with a significance level set at 0.05. The log-likelihood method was employed to estimate the parameters, and odds ratios were utilized to identify the influence of each predictor variable that best explained the occurrence of canine distemper. Additionally, death, age, neuter status, breed, and sex were tested to see if they had a normal distribution and then organized in a binomial model to test if the death was influenced by the other factors.

## 3. Results

Of 412 dogs with CNS disorders, 12.38% (51/412) were excluded due to suspected CDV infection without confirmatory testing, and one dog (0.24% (1/412)) was excluded due to incomplete records. A total of 360 cases met the inclusion criteria, with 4.72% (17/360) assigned to the DG and 95.28% (343/360) to the CG. Diagnoses in the DG were confirmed in 17.65% (3/17) of cases by immunochromatography, in 76.47% (13/17) by RT-PCR, and in 5.88% (1/17) by *Lentz corpuscles* identification. The estimated prevalence of CDV in the Neurology and Neurosurgery Service was 4.72% (17/360) in the observed period. Epidemiological data for both groups is summarized in Table 1 and Table 2. Complete records of animals can be found in Appendix A (Appendix A).

The frequency of neurological clinical signs is summarized in Table 3. The average interval between the onset of neurological signs and the day of presentation for veterinary care was 60 days (SD: 93 days; range: 1 to 270 days). The most common neurological sign in the DG was motor dysfunction (70.59%; 12/17). Non-ambulatory tetraparesis was observed in 66.67% (8/12) dogs, ambulatory tetraparesis in 16.66% (2/12), tetraplegia in 8.33% (1/12), and ambulatory paraparesis in 8.33% (1/12). The second most frequently observed sign was altered mental status (52.94%; 9/17). Among these, 77.78% (7/9) were classified as depressed, 11.11% (1/9) progressed from depressed to stuporous, and 11.11% (1/9) improved from alert to depressed. The third most common sign was seizures (41.18%; 7/17). Focal seizures accounted for 42.86% (3/7), while generalized seizures were observed in 57.14% (4/7).

Regarding the anatomical distribution of CNS lesions, a multifocal pattern was most frequently noted, occurring in 58.82% (10/17) of the cases. A focal pattern was observed in 35.30% (6/17), and in 5.88% (1/17), the neurologic examination was not performed due to the clinical conditions of the patient. Among the multifocal cases, 30% (3/10) were located at the forebrain and brainstem, 20% (2/10) in the undefined CNS region, 10% (1/10) in multiple spinal cord regions, 10% (1/10) in the brainstem and multiple spinal cord regions, 10% (1/10) in the forebrain and T3–L3 spinal segment, 10% (1/10) in the cerebellum and brainstem, and 10% (1/10) in the forebrain, brainstem and multiple spinal cord regions. Focal lesions were located at the forebrain in 83.33% (5/6) of the cases, whereas spinal cord localization was documented in 16.67% (1/6) of the DG dogs. Myelopathy localization in focal or multifocal cases was distributed as follows: 60% (3/5) involved multiple sites, 20% (1/5) were restricted to the T3–L3 spinal segment, and 20% (1/5) to the C1–C5 segment.

Extra-neural clinical signs were recorded in 94.12% (16/17) of the DG cases and are summarized in Table 3. Analysis of the temporal relationship between systemic and neurological signs revealed that extra-neural manifestations followed neurological signs in 37.50% (6/16) of the cases. These signs preceded the onset of neurological signs in 31.25% (5/16) and occurred concurrently in 12.50% (2/16) of the dogs. In the remaining animals (25%; 4/16), timeline data regarding the sequence of clinical signs were unavailable or incomplete. When extra-neural signs appeared after the onset of neurological signs, they were observed after a median of 31 days (SD: 27.62; range: 2 to 70 days). Conversely, when they preceded the neurological signs, the latter appeared after a mean of 64 days (SD: 36.46; range: 20 to 113 days).

The highest frequency of onset occurred in May, and autumn was the most common season (41.18%) (Figure 1). None of the cases involved dogs from the same owner. Among the Shih Tzu (*n* = 5) and Lhasa Apso dogs (*n* = 3), 12.5% (1/8) occurred in 2019, 50% (4/8) in 2020, 25% (2/8) in 2021, and 12.5% (1/8) in 2022. Notably, 50% (4/8) were diagnosed in May, two of them in 2020, and 75% (6/8) occurred during autumn, three of which were also in 2020. Regarding geographic distribution in the year with the highest number of cases, dogs were reported from different areas of the city (Central-West, West, and South), as well as from a neighboring municipality located near the southern region of the city where the hospital is located. No clustering pattern was observed in the temporal and spatial distribution.

Vaccination protocols were up-to-date in 35.29% (6/17) of DG dogs, outdated in 35.29% (6/17), and unavailable in 29.41% (5/17) of the cases. Epidemiological data and vaccine-related information are summarized in Table 4.

The overall mortality rate was 1.94% (8/412), while the lethality within the DG reached 47.06% (8/17). The median survival time for dogs in the DG was 754 days, as revealed by the Kaplan–Meier curve in Figure 2. The median survival time stratified by neurological phase is shown in Figure 3. Two animals were excluded from this analysis due to unavailable data on the date of neurological onset. Mortality had no significance when associated with sex, age, breed, or neutering status. Details of the animals that died during the study period are presented in Table 5.

## 4. Discussion

Our findings show that young adult dogs and specific breeds, namely Shih Tzu and Lhasa Apso, are at higher risk of developing neurological manifestations of CDV. Seasonal clustering was also evident, with a higher incidence of cases in autumn.

Although CDV is endemic in canine populations worldwide [7,23,24,25], few studies have focused on neurological manifestations and associated epidemiological risk factors. To the best of our knowledge, this is the first study to investigate the epidemiological characteristics and risk factors of neurologic CDV manifestations with a control group.

Our findings show a prevalence of neurological CDV of 4.72%, similar to previously reported CDV prevalence in general canine populations [13,16,20,25]. Despite a lethality rate of 47.06%, the overall mortality in our study (1.94%) was lower than previously reported in the same region [14], potentially due to the specialized neurological care available at our center. It is known that human neurology-focused centers can reduce patient stay, prevent worse outcomes, and lower mortality rates [26]. This underscores the benefit of targeted care in managing CDV-related complications.

Interestingly, although the median survival time was shorter in the acute neurological phase, the overall survival rate was lower in the chronic group. This indicates that chronic cases have prolonged yet poorer outcomes, despite the absence of a statistically significant difference (*p* = 0.95). Regardless of the neurological phase, seizures were the most common clinical sign associated with death or sequelae in the deceased dogs (62.50%; 5/8). This is consistent with the literature, which reports premature death [27] and a negative impact on the quality of life of both dogs and their owners [28] in cases of epilepsy. Furthermore, status epilepticus, observed in 37.5% (3/8) of these patients, is associated with a high mortality rate and complications, especially when treatment is delayed [29]. These findings suggest that seizures may be associated with a worse prognosis in CDV and could be more challenging to control, requiring more aggressive therapy and further research into specific treatments.

Notably, this study is the first to identify statistical associations between specific breeds and the likelihood of developing neurological CDV. Prior literature has often cited a high prevalence of CDV in mixed-breed dogs, likely reflecting broader environmental and socioeconomic factors rather than genetic susceptibility [8,14,16]. Our controlled comparison supports a genuine predisposition in Shih Tzu and Lhasa Apso. It is important to emphasize that the dogs in the DG group from these breeds did not originate from the same neighborhood, share the same owner, or attend the clinic during the same period, which significantly reduces the likelihood of an outbreak-related bias.

Conversely, all animals with an up-to-date vaccination protocol were either Shih Tzus or Lhasa Apsos, reinforcing the susceptibility of these breeds. It is well known that brachycephalic breeds are predisposed to upper respiratory tract (URT) disorders and often receive their first URT diagnosis at a younger age than non-brachycephalic high-risk breeds [30]. Therefore, these breeds may have reduced protection due to their anatomical conformation, which could predispose them to CDV, as the infection occurs through the inhalation of aerosols containing viral particles from the secretions of infected animals [2,21]. However, it remains unclear to what extent a brachycephalic conformation contributes to the increased risk of URT disorders [30]. Conversely, one study showed that brachycephalic breeds are less susceptible to developing CDV-induced encephalitis than dolichocephalic breeds, although this study lacked a control group [16]. Therefore, further studies are needed to elucidate the underlying reasons for these findings.

The DG dogs were younger than the CG dogs, confirming the age-related susceptibility to infection. In this study, all patients but one were under five years of age, aligning with previous studies that describe animals up to six years as being more predisposed [4,8,10,12]. However, one case involved a 10-year-old, reinforcing that age alone is insufficient to rule out CDV infection.

No significant associations were observed between neurological CDV and sex, weight, or neuter status, aligning with previous findings [14,24,31,32,33]. An earlier study noted a possible link between obesity and seropositivity for CDV [13], but this may reflect confounding factors such as immunologic memory or non-specific ELISA results.

CDV infections frequently induce multifocal lesions in the CNS [17,24] and appear to primarily target key brain cells [34]. In the present study, multifocal CNS involvement was the most common pattern, particularly affecting the forebrain and brainstem. These findings highlight the potential of CDV to cause lesions in multiple regions due to its marked brain tropism [35]. It is noteworthy that cerebellar signs were observed in only one patient, a finding also supported by a previous study [14]. However, it is important to recognize that lesions in this area may lead to subclinical signs [17]. Thus, the possibility of subclinical cerebellar involvement should not be excluded in positive cases [8,14].

Regarding clinical presentation, motor deficits were the most common neurological manifestation. Interestingly, myoclonus was less prevalent than in previous studies [14,17]. This is supported by previous findings where myoclonus was more frequently observed in dogs negative for distemper than in those positive for the disease [20], suggesting it should not be considered pathognomonic for CDV.

It is reported that respiratory, gastrointestinal, and dermatological signs typically appear within ten days of infection, when CDV spreads via cell-associated viremia to epithelial cells. CNS signs, however, usually begin later, around 20 days post-infection [6]. In our study, however, the timing of systemic compared to neurological signs varied, with no consistent progression identified. Interestingly, only one patient with neurological CDV did not exhibit systemic signs. This variation complicates early diagnosis and highlights the importance of a thorough clinical and neurological evaluation.

The seasonal clustering observed in autumn aligns with studies suggesting increased CDV stability in colder temperatures [33]. A plausible explanation for this pattern is that lower temperatures enhance the persistence and prolong the survival of CDV in the environment [13]. Additionally, cold weather may trigger stress-induced immunosuppression in young animals, increasing their susceptibility to infection [14].

It is known that vaccinated animals have a lower risk of developing infections compared to non-vaccinated ones [12]. However, there is an increasing discussion that fully vaccinated patients may become infected, suggesting vaccine failures. Previous studies identified that 10.87% to 16% of patients with distemper had an updated vaccination schedule [26]. In our research, the data are even more alarming, as over one-third of neurologically affected dogs had up-to-date vaccinations. Moreover, mortality was higher among dogs with up-to-date vaccinations than among those with outdated ones. This raises critical concerns regarding vaccine efficacy. Potential causes include antigenic mismatch, host-related factors, or waning immunity, potentially linked to antigenic drift among circulating wild-type CDV strains, or outdated vaccine formulations [3,36]. Unfortunately, detailed information about the vaccine brands used, and time elapsed between the last vaccine dose and the onset of clinical signs was not consistently available in medical records. Observing such information may be important in future prospective studies to assess whether potential decreased immunity could explain infections observed in vaccinated animals.

Previous studies have documented polymorphisms in circulating CDV strains that may impact vaccine efficacy [9]. Another hypothesis is that failures in vaccine storage, administration, or timing relative to the patient’s clinical status may have compromised the efficacy of immunization. The observation that a significant proportion of fully vaccinated animals had either not been vaccinated by veterinarians or lacked documentation on how vaccination was performed underscores the critical importance of ensuring that immunizations are conducted by qualified professionals, following proper protocols and after careful health evaluation of the patient.

Limitations of this study include its retrospective design, reliance on available medical records, and the exclusion of suspected cases without molecular confirmation. Nonetheless, our findings provide novel insights into the epidemiology of neurological CDV.

## 5. Conclusions

The findings of this study underscore that distemper remains a significant concern for canine health, given the high prevalence and mortality rates among affected animals. However, specialized neurological care may improve clinical outcomes. Furthermore, it identified key epidemiological characteristics and risk factors associated with neurological forms of CDV infection in dogs. Young age, as well as belonging to the Shih Tzu or Lhasa Apso breeds, were significantly associated with a higher likelihood of developing neurological signs due to CDV. The highest frequency of neurological cases occurred during autumn, suggesting possible seasonal influence. Our findings underscore the importance of breed- and season-specific preventive measures and advocate for further studies on strategies against CDV.

## Figures and Tables

**Figure 1 viruses-17-00820-f001:**
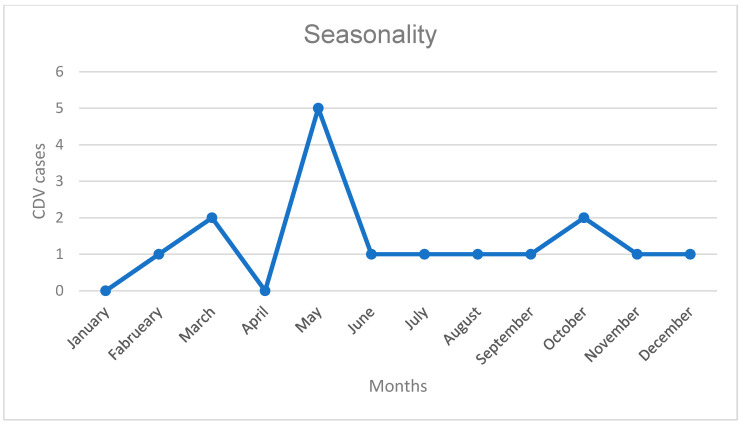
Distribution of CDV-related neurological signs onset by month. Caption: CDV: Canine Distemper Virus.

**Figure 2 viruses-17-00820-f002:**
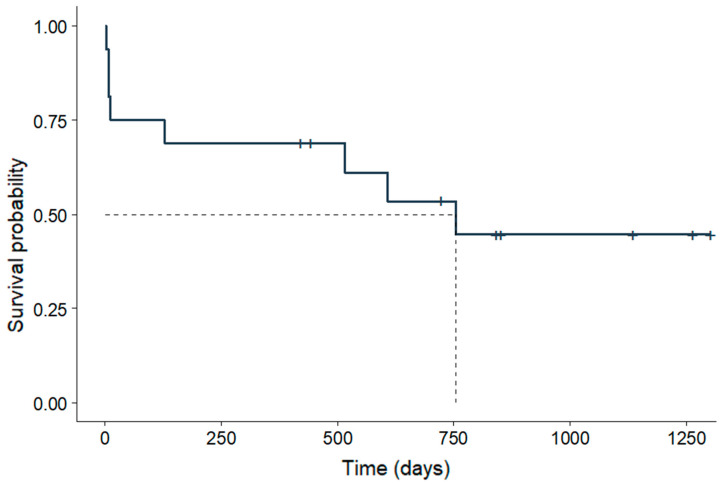
Kaplan–Meier survival analysis of 16 dogs diagnosed with CDV-related neurological disease. Caption: CDV: Canine Distemper Virus.

**Figure 3 viruses-17-00820-f003:**
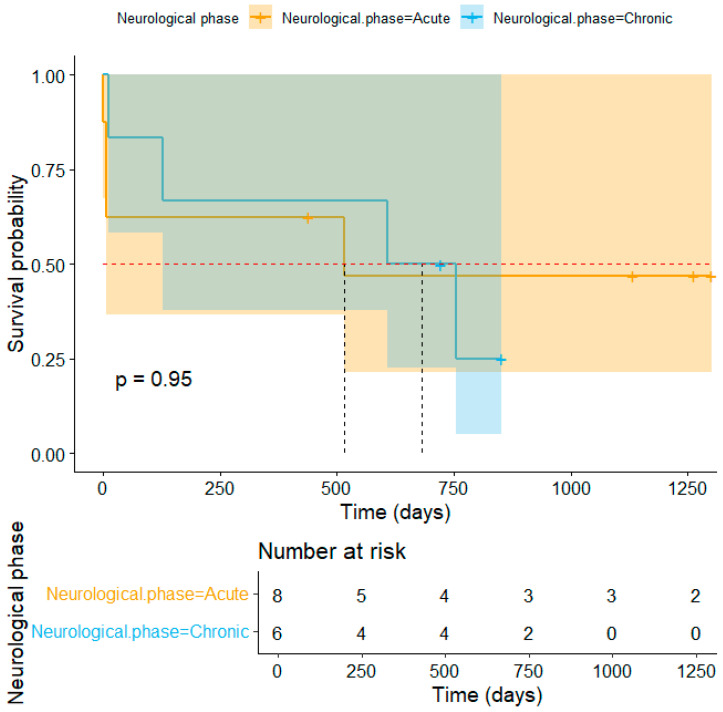
Kaplan–Meier survival curves of 14 dogs diagnosed with CDV-related neurological disease, stratified by neurological phase (Acute and Chronic). Dotted verticals represent median survival lines, and a dashed horizontal line marks the 50% survival threshold. Caption: CDV: Canine Distemper Virus.

**Table 1 viruses-17-00820-t001:** Epidemiological data for dogs diagnosed with CDV or other neurological disorders.

Risk Factor	DG (*n* = 17)	CG (*n* = 343)	*p*-Value
Breed	Pure = 10 Mixed-breed = 7	Pure = 201 Mixed-breed = 142	0.94338
Gender	Female = 8 Male = 9	Female = 181 Male = 162	0.39279
Neuter status	Yes = 4 No = 13	Yes = 134 No = 208 NI = 1	0.72536
Age	31 ± 30 months	68 ± 51 months	0.00690 **
Weight	6.4 ± 4.2 Kg	12.4 ± 11.8 Kg	0.08683

Note: CG: Control Group; DG: distemper group; NI: not informed; CDV: Canine Distemper Virus. ** *p* < 0.01.

**Table 2 viruses-17-00820-t002:** Breed distribution data for dogs diagnosed with CDV or other neurological disorders.

Breeds	DG (*n* = 17)	CG (*n* = 343)	*p*-Value	LOR	IC-LOR
Mixed breed	7	142	>0.05	−1.38053	[−1.81932, −0.96728]
Shih Tzu	5	37	0.00007 ***	1.53774	[0.76853, 2.29708]
Lhasa Apso	3	6	0.000264 ***	1.76084	[0.81950, 2.68406]
Pinscher	1	15	>0.05	−1.38053	[−1.81932, −0.96728]
Australian Cattle dog	1	0	>0.05	−1.38053	[−1.81932, −0.96728]
Others	0	143	>0.05	−1.38053	[−1.81932, −0.96728]

Note: CG: Control Group; DG: distemper group; CDV: Canine Distemper Virus; LOR: log odds ratios. *** *p* < 0.001.

**Table 3 viruses-17-00820-t003:** Frequency of neurological and extra-neural clinical signs in seventeen dogs diagnosed with CDV.

Types of Signs	Clinical Signs	Animals (*n*)	Percentage (%)
Neural signs (*n* = 17)	Paresis/plegia	12	70.59
	Altered mental state	9	52.94
	Seizure	7	41.18
	Postural change	6	35.29
	Tremors	6	35.29
	Myoclonus	6	35.29
	Ataxia	4	23.53
	Muscle hypotrophy	4	23.53
	Behavioral change	3	17.65
Extra-neural signs (*n* = 16)	Apathy	9	56.25
	Dehydration	7	43.75
	Ocular discharge	7	43.75
	Diarrhea and/or Vomiting	7	43.75
	Hyporexia/anorexia	6	37.50
	Hyperthermia	4	25.00
	Hyperkeratosis	3	18.75
	Respiratory signs	3	18.75
	Enamel hypoplasia	1	6.25
	Dermal pustules	1	6.25

Note: CDV: Canine Distemper Virus.

**Table 4 viruses-17-00820-t004:** Epidemiological data and vaccine-related information in positive canine distemper virus dogs.

Animal	Breed	Vaccination Status	Vaccine Brand	Performed by a Qualified Professional	Neurological Infection Phase	Month of Infection	Outcome
1	Mix breed	UN	UN	UN	Chronic	October	Dead
2	Shih Tzu	Up-to-date	Brazilian brand	No	Chronic	September	Dead
3	Lhasa Apso	Up-to-date	Zoetis	Veterinary	Acute	March	Alive
4	Mix breed	UN	UN	UN	Chronic	November	Dead
5	Lhasa Apso	Outdated	UN	UN	Chronic	June	Alive
6	Shih Tzu	Up-to-date	UN	Veterinary	Chronic	May	Alive
7	Mix breed	UN	UN	UN	UN	UN	Alive
8	Mix breed	Outdated	UN	UN	Acute	March	Dead
9	Shih Tzu	Up-to-date	Grascon	No	Acute	May	Dead
10	Mix breed	UN	UN	UN	Acute	July	Alive
11	Australian Cattle Dog	Outdated	UN	UN	UN	August	Alive
12	Mix breed	Outdated	Zoetis	UN	Acute	October	Alive
13	Pinscher	Outdated	UN	UN	Acute	February	Dead
14	Shih Tzu	Up-to-date	UN	UN	Acute	May	Dead
15	Mix breed	UN	UN	UN	Chronic	May	Dead
16	Lhasa Apso	Up-to-date	Zoetis	UN	Acute	May	Alive
17	Shih Tzu	Outdated	Zoetis	UN	Chronic	December	Alive

Note: UN: Unavailable.

**Table 5 viruses-17-00820-t005:** Canine distemper infection course characteristics in 8 deceased dogs.

Animal	Breed	Days from Presentation Until Death	Disease Phase at Death	Death Reason	Kind of Death	Sequelae
1	Mix breed	608	Chronic	Clinical deterioration	Euthanasia	Seizures
2	Shih Tzu	128	Chronic	Unavailable	Natural	Myoclonus and Seizures
4	Mix breed	754	Chronic	Status epilepticus	Natural	Seizures
8	Mix breed	6	Acute	Status epilepticus	Natural	-
9	Shih Tzu	7	Acute	Seizures and clinical deterioration	Euthanasia	-
13	Pinscher	1	Acute	Status epilepticus	Unavailable	-
14	Shih Tzu	516	Chronic	Unavailable	Natural	No
15	Mix breed	12	Acute	Clinical deterioration followed by cardiac arrest	Natural	-

## Data Availability

The data supporting the reported results of this study are not publicly available due to privacy and ethical restrictions. The data were collected from medical records of patients treated at a hospital and are owned by the respective animal owners and the institution. Access to the data can be granted upon request, subject to approval from the relevant ethical and legal authorities, in accordance with confidentiality agreements and privacy policies. For further information, please contact the corresponding author.

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
