# Peer review of "Neurological Manifestation of Canine Distemper Virus: Increased Risk in Young Shih Tzu and Lhasa Apso with Seasonal Prevalence in Autumn†"

_viruses, 2025, doi:10.3390/v17060820_

Round 1
Reviewer 1 Report
Comments and Suggestions for Authors
This manuscript addresses a relevant and timely subject by investigating the epidemiological characteristics and risk factors associated with neurological manifestations of canine distemper virus (CDV) infection. Utilizing a retrospective analysis of confirmed CDV cases, the study compares affected dogs to a control group with other neurological disorders. It highlights associations with specific breeds, age, and seasonality, contributing to a deeper understanding of susceptibility patterns and clinical presentations. However, the Introduction section would benefit from substantial structural and conceptual refinement to convey the rationale and specific aims of the study more effectively. Although the clinical relevance of CDV is acknowledged, the existing knowledge gaps and the justification for the present investigation are not adequately articulated. Additionally, while the primary objective of the study is not focused on vaccination, vaccine-related considerations are discussed in detail without sufficient empirical support. Overall, I recommend a major revision to address these concerns before the manuscript can be considered for publication.
Major comments:
Introduction
The manuscript mentions that the frequency of infected dogs associated with risk factors and clinical manifestations remains poorly characterized, but this point could be more clearly developed. Which specific risk factors are currently unclear? Expanding on what is known and still missing would better justify the study’s objectives.
Materials and Methods and Results
The number of cases in the CDV group (n=17) appears to be limited for multivariable statistical models. Could the authors clarify whether they performed any power analysis to determine if the sample size was sufficient to support the reported associations, particularly with respect to breed and age-related risk factors?
The manuscript presents p-values and odds ratios for several variables; however, confidence intervals are not provided. Including confidence intervals would allow readers to assess the precision and reliability of the effect estimates. Would the authors consider adding these to strengthen the statistical interpretation?
Discussion
The discussion highlights concern regarding vaccine failure, particularly about potential antigenic mismatch or waning immunity. However, the manuscript lacks critical details about the specific types or brands of CDV vaccines administered to the affected animals, as well as the number of dogs with confirmed vaccination histories. This information is crucial for evaluating the validity of any conclusions drawn about vaccine efficacy.
While vaccination status is mentioned (updated vs. outdated), the manuscript does not provide the time elapsed between the last vaccine dose and the onset of clinical signs. This information would be important to assess whether potential waning immunity could explain the observed infections in "vaccinated" animals.
Could the authors clarify which vaccines were used, provide data on the vaccination status of the animals included in the study, and specify the time elapsed between the last vaccine dose and the onset of clinical signs?
Without sequence data, it is difficult to assess whether antigenic mismatch, host-related factors, waning immunity, or antigenic drift among circulating wild-type CDV strains contributed to the clinical outcomes observed in vaccinated dogs.
Minor comments:
- The abbreviation "CNS" appears in lines 18 and 24 of the abstract without prior definition.
- Prevalence, mortality, and lethality metrics are reported at multiple points throughout the manuscript; however, the format of presentation is not fully consistent (e.g., sometimes expressed as percentages, elsewhere as proportions, or both). A standardized and uniform reporting format is recommended to enhance clarity and comparability.
- Statistical values in the manuscript are reported with varying levels of decimal precision (e.g., 1.9% vs. 1.94% or 47.06%). It is recommended that the level of precision throughout the text be standardized in accordance with the statistical methodology applied.
- Line 95 and 99: Replace “CD” with “CG” for consistency
- Line 155: I recommend including “ELISA” in the Abbreviations section.
- Line 163: Instead of "vs", use one of the expressions "as opposed to", "and", "compared to"
Reviewer 2 Report
Comments and Suggestions for Authors
General
The manuscript is an interesting article that describes new and useful epidemiological information of neurological clinical presentation of fifteen natural Canine Distemper Virus infected dogs from Brazil. The “Introduction” is clear and well-written, describing properly the aim of this epidemiological study. “Materials and Methods” includes all the needed details and complementary information to understand the statistical design, mainly how the authors recover the information from the clinical records of 5 years of the Veterinary Medical Teaching Hospital at the Federal University of Goiás. The “Results” section includes incomplete details of the study, although the main results are clearly exposed. In my opinion, the authors must include in this section, at least, a specific paragraph about the analysis of vaccinated and non-vaccinated dogs in the infected CDV group, because they describe this information in “Materials and Methods” and discused, but these details are not included as results. Furthermore, could be interesting to know if vaccinated and infected dogs were most frequent identify in any of studied breeds in CDV infected group. How many of vaccinated dogs died or were euthanized in comparision with non-vaccinated infected dogs? In the vaccinated infected dogs, how was the main clinical presentation? Acute or chronic? In the Control Group, the authors compare only the information about the breeds that were identified in the infected group, but they include in table 2, as “Others” breeds a group of 143 dogs. Please, include as an Appendix the complete distribution of breeds of all the dogs of Control Group, to compare with distribution of CDV-infected group. All these details could improve the better understanding of the results. If posible, comment any detail about the seasonality distribution of the infected cases of Shih-Tzu and Lhasa Apso dogs. Are these cases from the same owner, geographical region, or neighborhood? Were received in the same year as a part of an outbreak?
“Discusion” section have, at least, two details could be improved. The first, is linked with my comment of Results. They state in lines 168-169 “It is notable that over one-third of neurologically affected dogs had up-to-date vaccinations.”. Please include these details in the results, and rewrite the sentence in discusion in accordance with results. The second,
Finally, the “Conclusion” section include a paragraph of general informationnot linked with specific obtained results. Please, rewrite the paragraph, including only the conclusions of the obtained results.
In my opinion, the manuscript have potential to be accepted, but the described drawbacks must be addressed and resubmitted beforae final decisión.
Particular
Line 35: Substitute “Canine distempers …” by “Canine distemper …”
Line 112: Substitute “Caption: CDV: Canine Distemper Virus; * p<0,01” by “Caption: CDV: Canine Distemper Virus” (in this table you do not show statistical analysis details…)
Round 2
Reviewer 1 Report
Comments and Suggestions for Authors
I appreciate the authors’ thorough responses and the revisions made to the manuscript. All my previous concerns have been adequately addressed. I have no further comments and recommend the manuscript for publication in its current form.
Author Response
We sincerely thank you for your thoughtful comments and the positive evaluation of our manuscript. Your suggestions were truly enriching and helped us improve the quality of our work. We are very pleased with your final recommendation and hope the article will make a meaningful contribution to the field.
Reviewer 2 Report
Comments and Suggestions for Authors
In my opinion, and in accordance with the careful and fullfilled response of the authors to the referees suggestions, this new version, properly complies all the recommendations. My final recommendation is the acceptance in the present form.
God job!
Only any few gramar details to be fixed befor publication in the supplementary materials:
- In the first table, heading must be expanded to read de complete title of the heading (EPIDEMIOLOGICAL DATA'').
- In the heading of 2nd column of first table, probably you want to write "1st CONSULTATION DATE (DD.MM.YYYY)" instead of "1th CONSULTATION DATE (DD.MM.YYYY)"
- In the third table, substitute "SAZONALITY" by "SEASONALITY"
- In the same table, in the last row, only "DECEMBER" appear in capital. Please, fix.
- In the last table, please you must to expand the wide of the third column to allow the visualization of the complete sentence (for instance: "Seizures and clinical deteriora..." or "Clinical deterioration and car...")
Author Response
We sincerely thank you for your positive feedback and recommendation for acceptance. We are also grateful for your careful review of the supplementary materials.
We appreciate the detailed suggestions regarding the grammar and formatting issues, and we will make all the necessary corrections before publication:
-
The heading in the first table will be expanded to include the full title “EPIDEMIOLOGICAL DATA”.
-
The column header “1th CONSULTATION DATE” will be corrected to “1st CONSULTATION DATE”.
-
The term “SAZONALITY” will be replaced with “SEASONALITY”.
-
The formatting in the last row of the third table will be adjusted so that all month names follow the same capitalization.
-
The width of the third column in the last table will be expanded to ensure the full text is visible.
Thank you once again for your helpful comments and your kind words.
Round 3
Reviewer 2 Report
Comments and Suggestions for Authors
My final recommendation is the acceptance in the present form.